# The Combined Effect of Ambient Conditions and Diluting Salt on the Degradation of Picric Acid: An In Situ DRIFT Study

**DOI:** 10.3390/ma15176029

**Published:** 2022-09-01

**Authors:** Roberto Sanchirico, Luciana Lisi, Valeria Di Sarli

**Affiliations:** Istituto di Scienze e Tecnologie per l’Energia e la Mobilità Sostenibili (STEMS), Consiglio Nazionale delle Ricerche (CNR), Via Guglielmo Marconi 4, 80125 Napoli, Italy

**Keywords:** in situ DRIFT spectroscopy, energetic materials, aromatic nitro compounds, picric acid, decomposition, diluting salt, KBr, ZnSe

## Abstract

An unexpected promoting effect of KBr, used as a diluting salt, on the degradation of picric acid (PA) was observed during in situ diffuse reflectance infrared Fourier-transform (DRIFT) spectroscopy experiments performed here under accelerated ageing conditions—at 80 °C and under an inert or oxidative atmosphere. While the formation of potassium picrate was excluded, this promoting effect—which is undesired as it masks the possible effects of test conditions on the ageing process of the material—was assumed to favor a first step of the decomposition mechanism of PA, which involves the inter- or intramolecular transfer of hydrogen to the nitro group, and possibly proceeds up to the formation of an amino group. An alternative diluting salt, ZnSe, which is much less commonly used in infrared spectroscopy than KBr, was then proposed in order to avoid misleading interpretation of the results. ZnSe was found to act as a truly inert diluting salt, preventing the promoting effect of KBr. The much more chemically inert nature (towards PA) of ZnSe compared to KBr was also confirmed, at much higher temperatures than DRIFT experiments, by dynamic differential scanning calorimetry (DSC) runs carried out on pure PA (i.e., PA without salt) and PA/salt (ZnSe or KBr) solid mixtures.

## 1. Introduction

From a thermodynamic point of view, energetic materials are unstable substances. Their decomposition can occur slowly even at room temperature, although degradation phenomena for energetic materials with an activation energy of decomposition higher than 170 kJ mol^−1^—such as aromatic and aliphatic nitro compounds and aliphatic nitramines—are supposed to be detectable only after thousands of years if correctly stored [1].

Nitroarenes decompose according to different modes depending on temperature: high temperatures promote the rupture of the C-NO_2_ bond requiring the highest energy, whereas at lower temperatures, processes with lower activation energy and lower exothermicity occur [2,3]. The decomposition of most aromatic nitro compounds takes place according to autocatalytic kinetics, due to catalysis by reaction products [2]. In this study, picric acid (PA) was chosen as a representative of this class of substances.

In order to determine the effects of long-term storage and predict their shelf life, energetic materials are subjected to accelerated ageing tests at temperatures typically ranging from 40 °C to 80 °C, so as to reproduce conditions that simulate a degradation process lasting several years in a much shorter time period [4]. In addition to being affected by temperature, PA is also known to react with metals to form metallic picrates, which have caused serious explosive accidents [5]. Moreover, the removal of nitro groups from some organic explosives—such as PA, trinitrotoluene, and nitroglycerine—can be induced by thermal decomposition or, alternatively, by contact with a potassium or sodium hydroxide solution at room temperature [6].

Thermal analysis technologies—especially differential thermal analysis (DTA), differential scanning calorimetry (DSC), and thermogravimetric analysis (TG)—have been widely used to investigate the decomposition of energetic materials, sometimes in combination with Fourier-transform infrared (FTIR) spectroscopy or mass spectrometry (MS) for the identification of degradation products [7,8,9]. In contrast, in situ diffuse reflectance infrared Fourier-transform (DRIFT) spectroscopy has not yet been used to characterize the degradation process of these systems, although it allows the heating of samples under different atmospheres, and represents a powerful technique for studying thermal phenomena also affected by the reactive nature of contacting gases.

Standard IR analysis of PA is performed through the traditional potassium bromide (KBr) pellet method, which involves the preliminary grinding of the sample with KBr powder before pressing the resulting solid mixture [10,11]. However, two further methods of sample preparation have been used in IR experiments on energetic materials other than PA, consisting of evaporating the solvent after dispersing a solution of the sample onto an alkali salt substrate [12], or gently spreading the sample in a thin layer on a substrate with the flat end of a spatula [13]. In all cases, spectra were recorded at room temperature. KBr does not contain bands in the mid-IR region of the spectrum and, consequently, does not mask the bands of the IR spectrum of the investigated sample [14]. Likewise, the DRIFT technique also involves grinding the sample with KBr powder, although no subsequent pressing into a disk is performed.

As mentioned above, in in situ DRIFT experiments, heating of the sample and/or its exposure to reactive conditions (e.g., oxidative and/or wet atmosphere) is allowed. However, the possible interaction between PA and KBr under specific test conditions must be carefully analyzed and, if it occurs to an extent that masks the effect of the test conditions themselves, alternative solutions to KBr must be identified in order to avoid a misleading interpretation of results.

The possible degradation of PA under accelerated ageing conditions—at 80 °C and under an inert or oxidative atmosphere—was investigated here through in situ DRIFT. This study provides insight into the undesired effect of KBr as a diluting salt, causing a limited but detectable transformation of PA not assignable to the conditions selected to perform the accelerated ageing tests, and proposes the use of a truly inert diluting salt—zinc selenide (ZnSe)—as an alternative to KBr. A joint DSC study shows the different thermal behavior of PA/KBr and PA/ZnSe solid mixtures at much higher temperatures than DRIFT experiments.

## 2. Materials and Methods

Picric acid (PA) (purity ≥ 99%) was purchased from Sigma-Aldrich (St. Louis, MO, USA). For safety reasons, this material was supplied moistened with water. The weight percentage of water was ≥ 35%. Dry material, hereafter referred to as “fresh” PA, was prepared by subjecting the moistened material to vacuum-drying at room temperature. PA (always purchased from Sigma-Aldrich, and with purity ≥ 99%) stored as dry powder for more than 10 years at ambient conditions in the Calorimetry Laboratory of CNR-STEMS was also available. This “naturally aged” PA was selected as a reference aged material.

In situ DRIFT experiments were performed with a PerkinElmer Spectrum GX spectrometer (Beaconsfield, United Kingdom) at 4 cm^−1^ resolution, averaging each spectrum over 64 scans. In order to collect the spectrum of fresh PA, fresh material was diluted in KBr (IR-grade potassium bromide purchased from Sigma-Aldrich), which is the most commonly used diluting and background material in IR spectroscopy. Then, 2 wt.% PA/KBr was placed in a PIKE DRIFT accessory (Fitchburg, WI, USA) equipped with a heat chamber and a ZnSe window. The PA/KBr solid mixture was treated at 30 °C for 1 h under an inert flow (Ar, 20 cc min^−1^), and then a DRIFT spectrum was recorded and ratioed against the spectrum of pure KBr. In another DRIFT experiment, 2 wt.% (fresh) PA was diluted in ZnSe (optical-grade zinc selenide purchased from Sigma-Aldrich), which was also used to record the background spectrum in this case. The same pretreatment as the PA/KBr mixture was carried out on the PA/ZnSe mixture. Spectra were also recorded during experiments under “accelerated ageing” conditions, as detailed in Table 1.

The temperature of 80 °C was chosen as being sufficiently lower than the melting point of PA (about 120 °C) to prevent liquefaction. In these cases, spectra were ratioed against the spectrum of the pure diluent (KBr or ZnSe) recorded at 80 °C.

Further DRIFT spectra were recorded for naturally aged PA, PA thermally treated (i.e., artificially aged) as pure powder (i.e., without diluting salt) ex situ (in a closed glass tube) at 80 °C for 24 h in air using a dry bath heater (THERMOBLOCK TD 200 P2+ by FALC Instruments, Treviglio, Italy), and potassium picrate (KP). In these experiments, 2 wt.% PA or KP was diluted in KBr, and spectra were collected at 30 °C under an Ar flow (as described for fresh PA). KP was prepared starting from an aqueous solution of PA and slowly adding K_2_CO_3_ up to a pH value of 7.5 [15]. The precipitate was filtered, washed with double-distilled water and, finally, dried at room temperature. Spectra were also collected at increasing times of exposure (up to 6 h) of 2 wt.% KP/KBr to 80 °C under an Ar flow.

DSC analysis was carried out with a PerkinElmer DSC 8000 instrument (Shelton, CT, USA) equipped with an Intracooler II cooling system. For each test, a few milligrams (0.5–2.0 mg) of material were loaded into a 30 μL stainless steel pan operating at a maximum working pressure of 150 bar and in the −170–400 °C temperature range. Dynamic DSC experiments were performed at a heating rate of 20 °C min^−1^ (exploring the 50–450 °C temperature range) on fresh and naturally aged PA, pure KBr, pure ZnSe, and PA/salt (KBr or ZnSe) solid mixtures with 50 wt.% PA. This much higher weight percentage of PA compared to the DRIFT experiments was related to the very small amount of sample that could be loaded into the pans used for the DSC experiments. Using 2 wt.% PA in the solid mixtures, as in the DRIFT experiments, would have represented too low an absolute amount of PA to be weighed without too large an experimental error.

## 3. Results and Discussion

### 3.1. Fresh Picric Acid (PA)

In Figure 1, the typical signals of fresh PA can be observed in the spectrum of 2 wt.% PA/KBr collected at 30 °C under an inert Ar flow, with the appearance and sometimes overlapping of many bands that are not unambiguously attributed. The bands at 1550 cm^−1^ and 1350 cm^−1^ are associated with the asymmetric and symmetric stretching of NO_2_, respectively [10,16]. As expected, these bands are shifted towards lower wavenumbers compared to nitroalkanes because they are attached to the aromatic ring. The bands at 1530 cm^−1^ and 1435 cm^−1^ are associated with stretching vibrations of aromatics [10], while the signal at 1310 cm^−1^, partially superimposed on the band at 1350 cm^−1^, is associated with the C-N bond [17]. On the other hand, the sharp band at 3104 cm^−1^ is assigned to the stretching of O-H [11] or C-H [17], whereas the small peak at 3250 cm^−1^ is unambiguously attributable to hydroxyl groups, likely together with that at 3300 cm^−1^. The broad band in the 3650–3000 cm^−1^ region is a combination of hydroxyls of PA and adsorbed water. The bands at 1275 cm^−1^ and 1150 cm^−1^ are assigned to the C-O bond [11,17], although 1277 cm^−1^ is also the frequency assigned to the stretching [10,17] and rocking [17] of C-N. The band at 1630 cm^−1^ is attributed to stretching vibrations of aromatic C=C, whereas the band at 1090 cm^−1^ is attributed to C-H out-of-plane bending [10].

### 3.2. Thermally Treated PA

#### 3.2.1. Diluting Salt: KBr

In Figure 2, the spectrum from Figure 1 is shown along with the spectra of the same system treated at 80 °C for up to 48 h under an inert flow. Specifically, Figure 2a shows the spectra in the 3500–2500 cm^−1^ region, whereas Figure 2b shows the spectra in the 1800–1000 cm^−1^ region. Although these spectra appear very similar, some differences can be observed, which could be tentatively attributed to a partial modification of PA accelerated by the high temperature. The most remarkable difference induced by the thermal treatment is the appearance of a new band centered at 1485 cm^−1^, whose intensity increases with the time of exposure to 80 °C. Nevertheless, other less evident differences can be noticed—a slight shift of the sharp band at 3104 cm^−1^ towards lower frequencies (Figure 2a), along with the shift of asymmetric and symmetric N-O stretching from 1550 cm^−1^ to higher frequencies and from 1350 cm^−1^ to lower frequencies, respectively (Figure 2b). Other bands present in the spectrum of fresh PA, such as the double peak at 3300–3250 cm^−1^, are better defined with increasing time of exposure to high temperature, likely due to the loss in intensity of the large broad band of the hydroxyl of adsorbed water which, in the case of the more hydrated fresh sample, partially masks these bands (Figure 2a). However, the attribution of both the new band at 1485 cm^−1^ and the shift of other pre-existing bands is not trivial.

At temperatures over which the thermal stability of nitroarenes is normally studied, three modes of decomposition are postulated, whose relative dominance changes with temperature [2,3]: (a) homolysis of the C-NO_2_ bond—this is a high-energy event (requiring about 300 kJ mol^−1^) and, therefore, can occur only at high temperatures; (b) intermolecular (from another nitroarene) or intramolecular (from another group on the same arene ring) transfer of hydrogen to the nitro group, resulting in the loss of HONO—this transfer requires about half the energy of the homolysis; and (c) nitro/nitrite isomerization. According to these possible events, some mechanisms could be assumed to be responsible for the changes observed in the DRIFT spectra of artificially aged (i.e., thermally treated) PA (Figure 2). Since the -NO_2_ bands (i.e., asymmetric and symmetric N-O stretching) shift slightly but do not disappear, the homolysis of the C-NO_2_ bond is hardly feasible. Other transformations of PA should be hypothesized, such as the inter- or intramolecular transfer of hydrogen to the nitro group. This transfer, which requires less energy, and can be assumed to occur even at low temperatures (e.g., 80 °C), could be responsible for possible subsequent molecular changes. In the case of PA, the intramolecular transfer of hydrogen can obviously only take place on one of the two ortho-nitro groups [18].

When the nitro group interacts with the hydrogen of a hydroxyl group, the bands of asymmetric and symmetric stretching shift towards higher and lower frequencies, respectively [16]. The opposite shifts of asymmetric and symmetric N-O stretching shown in Figure 2b can therefore account for an interaction of the nitro group with the hydrogen of the hydroxyl group of the same molecule or of another molecule of PA. If the sharp band at 3104 cm^−1^ is attributed to the hydroxyl of PA, the shift observed for this band can actually be assigned to the transfer of hydrogen, although the extent of this shift is quite low. Indeed, much higher shifts are expected for an interaction of OH with the nitro group [19]. As a consequence, the occurrence of the transfer of hydrogen to the nitro group is supported only by the shift of the -NO_2_ bands. On the other hand, if the band of the hydroxyl of PA is included in the large broad band observed in the 3650–3000 cm^−1^ region, its possible shift is hardly detectable.

The explanations given so far do not justify the formation of a completely new band—namely, the band at 1485 cm^−1^. This signal is typically assigned to the C=C stretching of substituted aromatics (such as toluene, diethylbenezene, etc.), and could suggest that some modification of the aromatic ring occurs due to the different nature of the functional groups, as in the case of the amino group [20]. The associated great increase in the sharp band at 1370 cm^−1^—present as a barely detectable signal in the case of fresh PA, and typically assigned to secondary or tertiary amino group bound to an aromatic ring [20]—could confirm this hypothesis. On the other hand, the formation of an amino group from a nitro group through the initial transfer of hydrogen has been reported for trinitrotoluene [3], and can easily be assumed for PA as well.

In conclusion, although not unambiguously defined, a limited but detectable modification of PA—possibly related to the intra- or intermolecular transfer of hydrogen to the nitro group, and to the formation of an amino group—takes place when keeping the PA/KBr sample at 80 °C only for a few hours under an inert atmosphere. This is undoubtedly an unexpected result for a highly stable compound such as PA (see, e.g., Ref. [1]).

When comparing the spectra of Figure 2 with those of Figure 3, it appears that the formation and shift of bands are accelerated by the presence of O_2_. Specifically, bands at 1485 cm^−1^ with almost the same intensity appear after a 48 h treatment under an Ar flow (Figure 2b) and a 4 h treatment under an O_2_/Ar flow (Figure 3b).

Similar effects can also be observed in the case of naturally aged PA, i.e., PA stored for more than 10 years at ambient conditions in the Calorimetry Laboratory of CNR-STEMS (Figure 4). A band at 1485 cm^−1^ is also present in this case (in addition to a band at 1565 cm^−1^) (Figure 4b), whereas the possible dominance of a double band at 3300–3250 cm^−1^ is likely masked by the stronger hydration of aged PA, resulting in a larger broad band in the region of hydroxyl stretching (Figure 4a).

The results presented above were crosschecked by comparing the spectrum of fresh PA with that of PA aged, as pure powder (i.e., without KBr), ex situ at 80 °C for 24 h in air (Figure 5). No significant differences can be observed between the two spectra. Notably, the band at 1485 cm^−1^ was not formed when the ageing treatment was performed ex situ on pure PA. A combined effect of high temperature and the presence of KBr is therefore likely responsible for the modification of signals observed in Figure 2 and Figure 3, whereas the ex situ ageing treatment of pure PA at 80 °C for 24 h in air (i.e., under about 20 vol.% O_2_ rather than 5 vol.% O_2_ as in the in situ DRIFT experiments shown in Figure 3) is not sufficient to modify the molecular structure of the substance under examination. In this regard, Figure 6 highlights the high thermal stability of PA, showing the comparison between the DSC peaks of fresh and naturally aged material collected during dynamic runs. The effect of more than 10 years of storage at ambient conditions is rather limited. It can be detected in the temperature range of 270–300 °C, where the small shoulder of the peak of naturally aged PA suggests that, during the storage, the material undergoes a transformation that initiates the decomposition of a small fraction of the sample at slightly lower temperatures than fresh PA.

In conclusion, KBr does not behave as an inert diluting salt, but rather affects the simulated ageing of PA which, therefore, cannot be unambiguously attributed to the test conditions. Due to the large amount of KBr in the PA/KBr solid mixture, a close contact between KBr and PA is established. On the basis of this close contact, a promoting effect of KBr on the degradation of PA, as also reported for the decomposition of potassium picrate [15], or even the formation of potassium picrate via a solid/solid reaction, as warned by Coates [20], can be assumed, with both events being possibly favored at 80 °C. The stability of metal picrates with respect to PA is debated. Ju et al. [17] reported that potassium picrate is more prone to decomposition than PA. Some C-C bonds in potassium picrate are very weak, suggesting that they could be ruptured simultaneously with the C-N bond in the initial decomposition process. This could support the hypothesis of the formation of potassium picrate in the DRIFT cell, followed by an easier degradation of the new compound, although no signals attributable to the rupture of the C-NO_2_ bond can be observed in Figure 2 and Figure 3. In contrast, as shown by the DSC results reported by Matsukawa et al. [5], the decomposition of alkali metal picrates begins at higher temperatures than that of PA, suggesting a higher thermal stability of the salts.

In order to understand whether, at 80 °C, potassium picrate (KP) is formed, which then degrades, or if KBr promotes the decomposition process of PA, KP was prepared as described in Section 2, and a 2 wt.% KP/KBr mixture was loaded into the DRIFT cell. In Figure 7, the DRIFT spectrum of fresh KP is shown along with that of fresh PA.

As predicted by Ju et al. [17] using the density functional theory (DFT) method, the spectrum of KP shows largely the same bands as that of PA, excluding the broad band associated with the stretching mode of OH, which is obviously absent in the salt, whilst the sharp band at 3104 cm^−1^ is preserved, although slightly shifted towards lower wavenumbers. This can help to disambiguate the attribution of this signal to the stretching of C-H or O-H. Indeed, the unmodified presence of the sharp band in the spectrum of KP definitely indicates that this band is associated with the stretching of C-H in the aromatic ring, with OH being absent in the potassium salt. Last but not least, the new band at 1485 cm^−1^ observed for thermally treated PA (Figure 2 and Figure 3) is not detectable in the case of KP. This rules out the in situ formation of KP, and suggests that the degradation process of PA is favored at high temperatures by the presence of KBr. On the other hand, unlike PA, KP is thermally stable, as confirmed by Figure 8, where the spectrum of fresh KP is shown along with the spectra recorded at increasing times of exposure of this material to 80 °C.

The similarity of the spectra in Figure 8 provides two key results: KP is more stable than PA, and the hydrogen of the hydroxyl group of PA is involved in the interaction with the vicinal nitro group or a nitro group of another molecule of PA, promoting the first step of the degradation mechanism of this nitroarene. In contrast, the hydroxyl group is absent in the case of KP, which therefore exhibits higher stability.

#### 3.2.2. Diluting Salt: ZnSe (Versus KBr)

KBr exerts a promoting effect on the degradation of PA and, therefore, is an unsuitable diluting salt for investigating its accelerated ageing via in situ DRIFT experiments. In an attempt to identify an effective alternative solution, DRIFT experiments were repeated using ZnSe (instead of KBr) powder as the diluting salt. Recall that the window of the in situ DRIFT cell used in this work was made of ZnSe, which is much more water-tolerant and thermally resistant than KBr. The same concentration (2 wt.%) of (fresh) PA was diluted in ZnSe powder. As with KBr, the spectra of PA diluted in ZnSe were ratioed against pure ZnSe. Figure 9 shows the DRIFT spectra of 2 wt.% PA/ZnSe. The blue curve represents the fresh sample, whereas the other curves represent the sample artificially aged at 80 °C for up to 24 h under a 5 vol.% O_2_/Ar flow.

Even the spectrum of PA treated at 80 °C for 24 h under an oxidative atmosphere does not show significant differences compared to the spectrum of fresh PA, indicating the chemically inert behavior of ZnSe. This is clearer in Figure 10, which shows the spectra of 2 wt.% PA diluted in KBr or ZnSe after 24 h of treatment at 80 °C under a 5 vol.% O_2_/Ar flow. The band centered at 1485 cm^−1^ is present in the case of PA/KBr, but totally absent in the case of PA/ZnSe.

The much more chemically inert nature (towards PA) of ZnSe compared to KBr was also confirmed, at much higher temperatures than in the DRIFT experiments, by dynamic DSC runs carried out on pure PA (i.e., PA without salt) and PA/salt (ZnSe or KBr) solid mixtures. For these runs, Figure 11 shows the specific heat power as a function of temperature, whereas Table 2 gives the corresponding heat of reaction. It is worth mentioning that dynamic DSC runs were also carried out on both pure KBr and pure ZnSe (for the sake of brevity, these results are not shown here), but no thermal events were recorded over the temperature range shown in Figure 11.

The heat power curve collected for PA/ZnSe shows two relative maxima, with the second one (at higher temperature) located very close to the maximum of pure PA. The broadening of the peak as well as the presence of two local maxima can be attributed to a thermal dilution effect of the salt (i.e., to the higher thermal inertia of the PA/ZnSe system compared to pure PA). This is corroborated by the fact that, for PA/ZnSe, the heat of reaction referring to the mass of PA is almost the same as that of pure PA. In contrast, for PA/KBr, a simple thermal dilution effect cannot be invoked. In this case, the thermal process also exhibits a multistep behavior, but the heat of reaction referring to the mass of PA is much lower than that of pure PA.

In conclusion, unlike KBr, ZnSe does not promote the degradation of PA, acting as a truly inert diluting salt. The use of ZnSe to dilute PA is thus strongly recommended in order to avoid an incorrect interpretation of possibly detectable modifications in DRIFT spectra recorded during accelerated ageing tests on this material. In principle, this recommendation can also be extended to all energetic materials containing polar functional groups. When thermally stressed, such materials could more easily interact with KBr.

## 4. Conclusions

The degradation of picric acid (PA) was investigated under accelerated ageing conditions using in situ diffuse reflectance infrared Fourier-transform (DRIFT) spectroscopy. The results show that even the most stressful conditions tested—80 °C and the presence of O_2_ (5 vol.%)—do not cause a detectable degradation of PA within one day, provided that the material is diluted in a suitable salt. KBr, which is the most commonly used diluting salt in IR applications, exerts an undesired promoting effect on the degradation process of PA, causing a “false” ageing that is not attributable to the test conditions. Specifically, it favors a first step of the decomposition mechanism of PA, which involves the transfer of hydrogen from the hydroxyl group to the vicinal nitro group (intramolecular transfer) or to the nitro group of another molecule (intermolecular transfer), and possibly proceeds up to the formation of an amino group. In contrast, ZnSe acts as a truly diluting salt towards PA, exhibiting a much more chemically inert nature than KBr, as also confirmed—at much higher temperatures than in the DRIFT experiments—by differential scanning calorimetry (DSC) runs carried out on pure PA (i.e., PA without salt) and PA/salt (ZnSe or KBr) solid mixtures. The use of ZnSe to dilute PA is thus strongly recommended in order to avoid an incorrect attribution of possibly detectable modifications in DRIFT spectra recorded during accelerated ageing tests on this material. Aromatic nitro compounds other than PA, such as those not containing a hydroxyl group, could in principle not give the same interaction with KBr. However, this must be carefully verified in order to provide a correct interpretation of the phenomena observed via in situ DRIFT.

## Figures and Tables

**Figure 1 materials-15-06029-f001:**
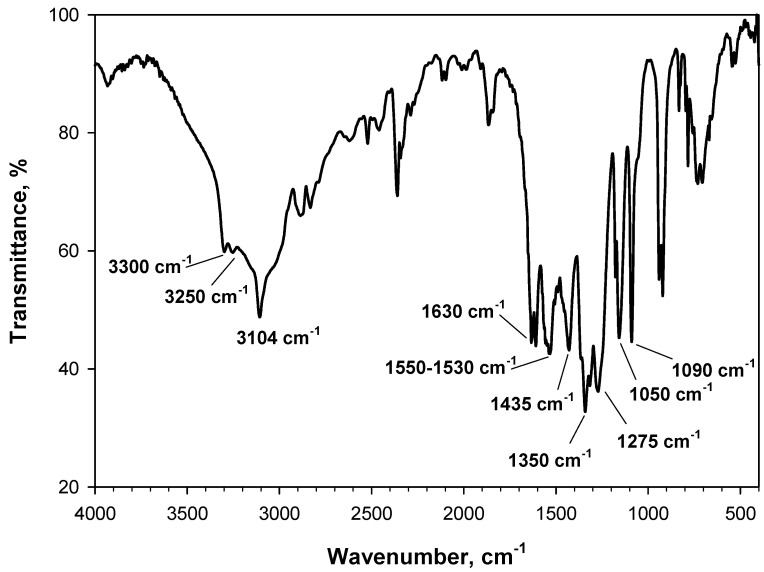
DRIFT spectrum of fresh PA diluted in KBr (2 wt.% PA/KBr) recorded at 30 °C under an Ar flow. The main typical signals of PA are labelled.

**Figure 2 materials-15-06029-f002:**
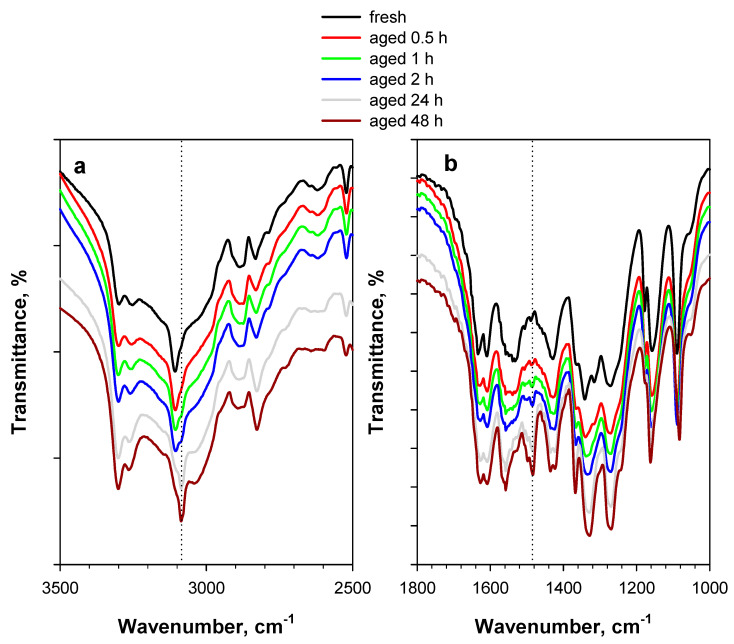
DRIFT spectra of 2 wt.% PA/KBr in (**a**) the 3500–2500 cm^−1^ region and (**b**) the 1800–1000 cm^−1^ region. The black curve represents the fresh sample, whereas the colored curves represent the sample artificially aged at 80 °C for up to 48 h under an Ar flow.

**Figure 3 materials-15-06029-f003:**
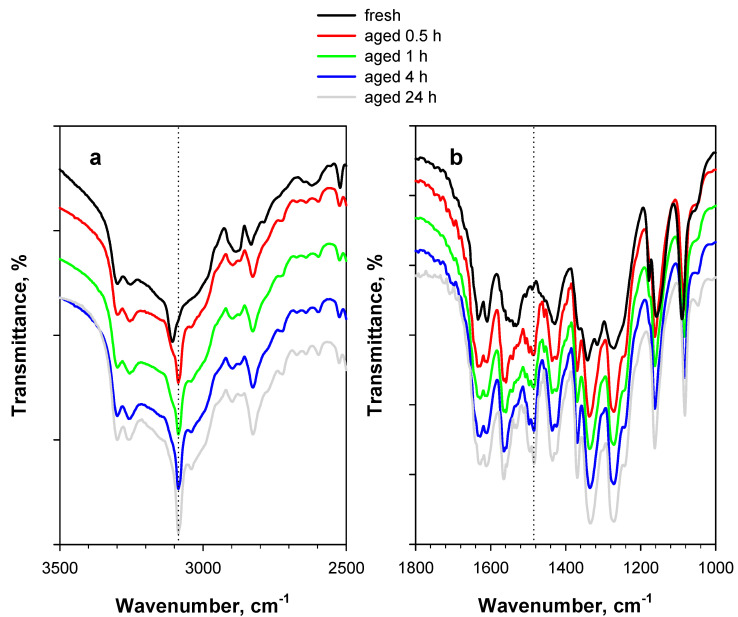
DRIFT spectra of 2 wt.% PA/KBr in (**a**) the 3500–2500 cm^−1^ region and (**b**) the 1800–1000 cm^−1^ region. The black curve represents the fresh sample, whereas the colored curves represent the sample artificially aged at 80 °C for up to 24 h under a 5 vol.% O_2_/Ar flow.

**Figure 4 materials-15-06029-f004:**
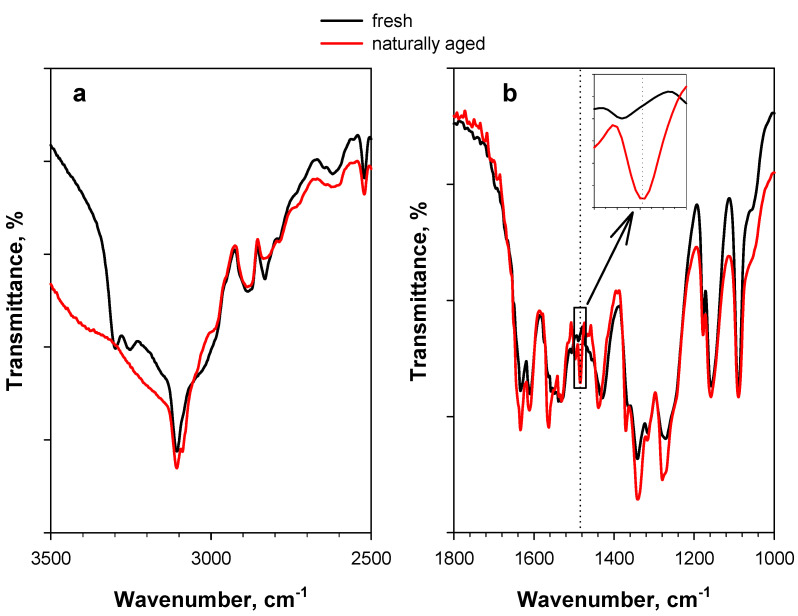
DRIFT spectra of 2 wt.% PA/KBr in (**a**) the 3500–2500 cm^−1^ region and (**b**) the 1800–1000 cm^−1^ region. The black curve represents the fresh sample, whereas the red curve represents the naturally aged sample (i.e., PA stored for more than 10 years at ambient conditions in the Calorimetry Laboratory of CNR-STEMS). Like the spectrum of fresh PA, the spectrum of naturally aged PA was also recorded at 30 °C under an Ar flow.

**Figure 5 materials-15-06029-f005:**
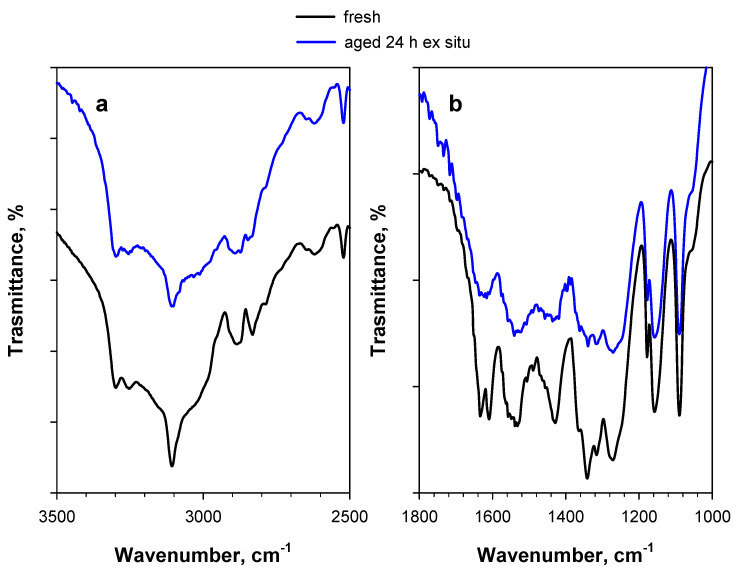
DRIFT spectra of 2 wt.% PA/KBr in (**a**) the 3500–2500 cm^−1^ region and (**b**) the 1800–1000 cm^−1^ region. The black curve represents the fresh sample, whereas the blue curve represents the sample artificially aged ex situ at 80 °C for 24 h in air. Like the spectrum of fresh PA, the spectrum of ex situ aged PA was also recorded at 30 °C under an Ar flow.

**Figure 6 materials-15-06029-f006:**
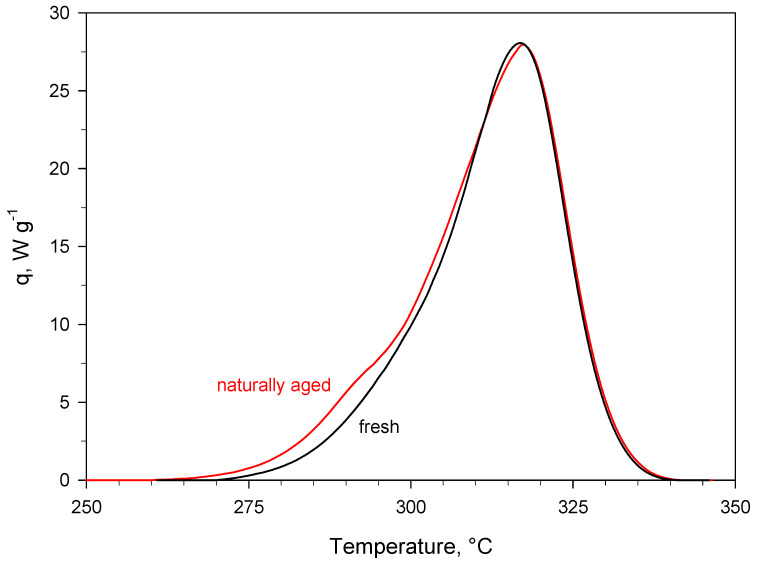
Specific (i.e., referring to the mass of sample) heat power curves (exothermal events) collected during dynamic DSC runs (heating rate of 20 °C min^−1^) carried out on fresh PA and naturally aged PA (i.e., PA stored for more than 10 years at ambient conditions in the Calorimetry Laboratory of CNR-STEMS).

**Figure 7 materials-15-06029-f007:**
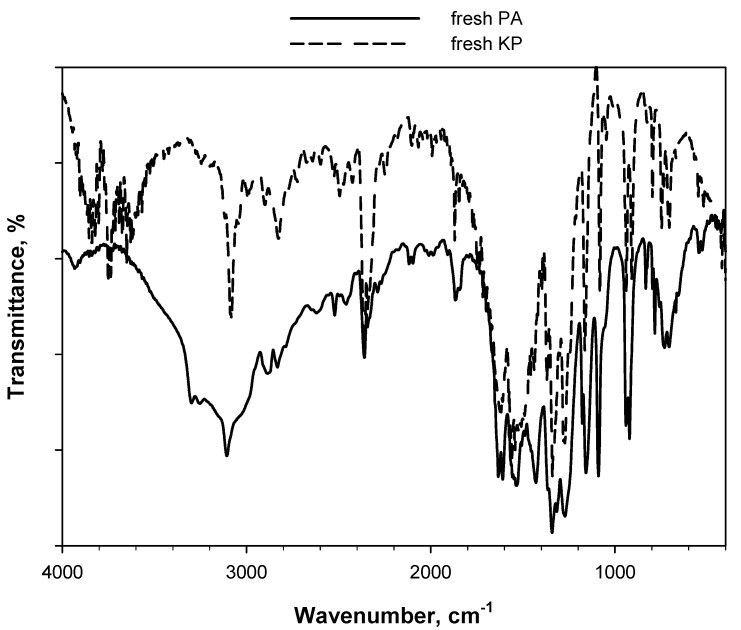
DRIFT spectra of fresh PA and fresh KP diluted in KBr (2 wt.% PA/KBr and 2 wt.% KP/KBr, respectively) recorded at 30 °C under an Ar flow.

**Figure 8 materials-15-06029-f008:**
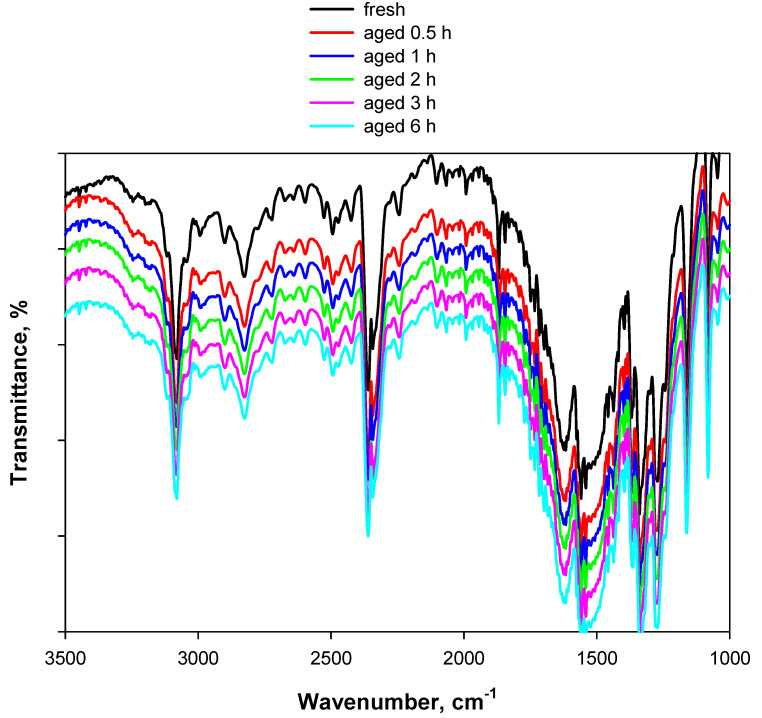
DRIFT spectra of 2 wt.% KP/KBr. The black curve represents the fresh sample (this spectrum was recorded at 30 °C under an Ar flow), whereas the colored curves represent the sample artificially aged at 80 °C for up to 6 h under an Ar flow.

**Figure 9 materials-15-06029-f009:**
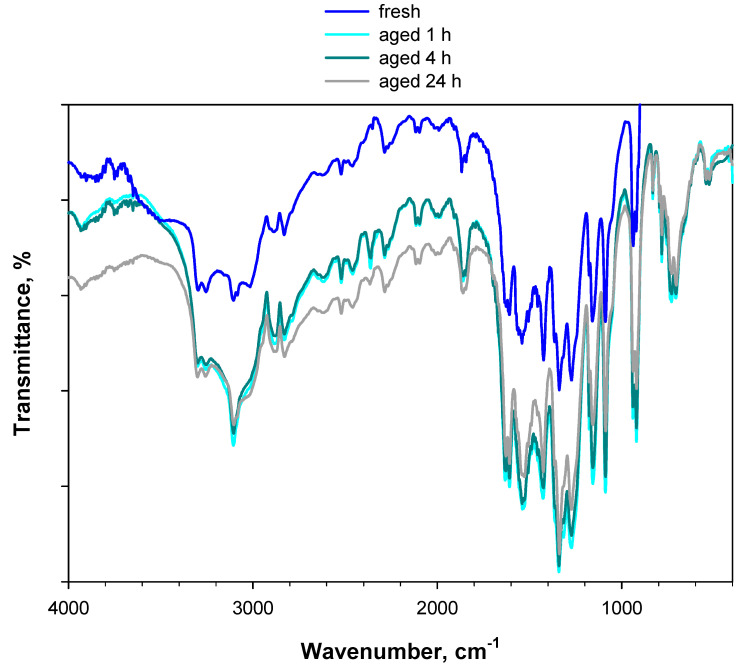
DRIFT spectra of 2 wt.% PA/ZnSe. The blue curve represents the fresh sample (this spectrum was recorded at 30 °C under an Ar flow), whereas the other curves represent the sample artificially aged at 80 °C for up to 24 h under a 5 vol.% O_2_/Ar flow.

**Figure 10 materials-15-06029-f010:**
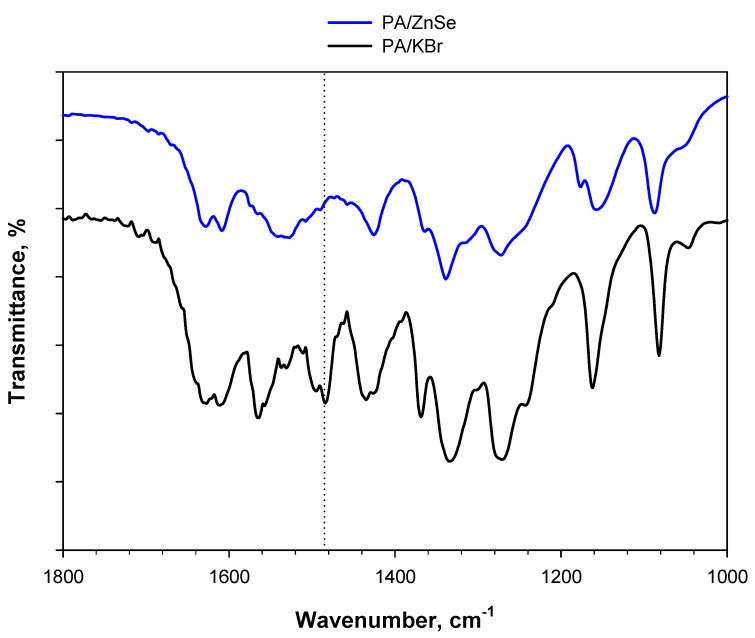
DRIFT spectra of 2 wt.% PA/KBr and 2 wt.% PA/ZnSe, both treated at 80 °C for 24 h under a 5 vol.% O_2_/Ar flow.

**Figure 11 materials-15-06029-f011:**
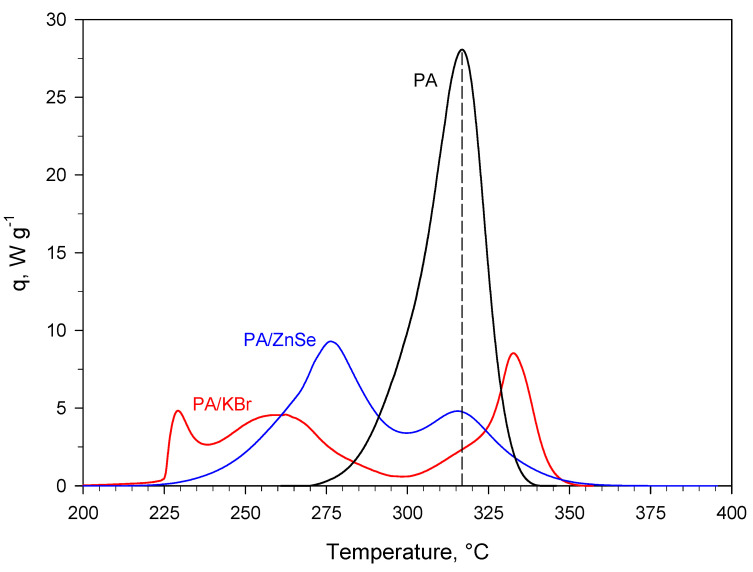
Specific (i.e., referring to the mass of sample) heat power curves (exothermal events) collected during dynamic DSC runs (heating rate of 20 °C min^−1^) carried out on pure PA (i.e., PA without salt) and PA mixed with KBr (50 wt.% PA/KBr) or ZnSe (50 wt.% PA/ZnSe).

**Table 1 materials-15-06029-t001:** Accelerated ageing: conditions of in situ DRIFT experiments.

Solid Mixture	Temperature, °C	Time of Exposure, h	Atmosphere
2 wt.% PA/KBr	80	(up to) 48	Ar flow
2 wt.% PA/KBr	80	(up to) 24	5 vol.% O_2_/Ar flow
2 wt.% PA/ZnSe	80	(up to) 24	5 vol.% O_2_/Ar flow

**Table 2 materials-15-06029-t002:** Heat of reaction corresponding to the curves shown in Figure 11.

System	ΔH_R_ Referred to m_TOT_ ^1^, J g^−1^	ΔH_R_ Referred to m_PA_ ^2^, J g^−1^
Pure PA	−3382	−3382
50 wt.% PA/KBr	−1011	−2022
50 wt.% PA/ZnSe	−1791	−3582

^1^ m_TOT_ is the total mass of (PA + KBr) or (PA + ZnSe). ^2^ m_PA_ is the mass of PA.

## Data Availability

The data presented in this study are available upon request from the authors.

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
