# Peer review of "The Combined Effect of Ambient Conditions and Diluting Salt on the Degradation of Picric Acid: An In Situ DRIFT Study"

_materials, 2022, doi:10.3390/ma15176029_

Round 1

Reviewer 1 Report

The authors present an experimental study of picric acid degradation. The work is interesting and include an appropriate instrumental characterization. Thus, I think the conclusions are robust and valid. I suggest publication of the manuscript after a minor revision including only two comments:

1.     It is difficult to perceive the text without a visual representation of the degradation reaction. I suggest the authors including a chemical scheme of the degradation and possible detectable products indicated.

2.     I suggest including some comment on the significance of the obtained results in practical field of energetic materials.

Reviewer 2 Report

Reference paper: materials-1865569

Thanks to the editor for inviting me to review the manuscript entitled The Combined Effect of Ambient Conditions and Diluting Salt on the Degradation of Picric Acid: An In Situ DRIFT Study.

The authors studied the potential effect of accelerated aging conditions on the degradation process of picric acid using in situ diffuse reflectance infrared Fourier transform spectroscopy. The results have high reference value for related research and application. Therefore, I recommend accepting it after major revision.

Comments:

1- The importance of the investigation should be justified and highlighted in the introduction.

2- The presentations of the introduction and other discussions are quite poor.

3- The novelty of the present work needs to be highlighted by the authors and I think it is not clearly presented in the manuscript.

3- Language should be thoroughly revised as several of the sentences are confusing and many errors can be found. I recommend using a professional editing service for the next version of the paper.

4- Abstract should be improved to well reflect the content.

5- A graphical abstract should be added.

6- Acronyms should be defined during their first use.

7- The discussion of the results needs a deep improvement in order to improve the quality of the manuscript.

8- The applicability of the present study should be addressed.

10- The quality and resolution of all figures should be improved.

11- Some important characterizations including TGA and SEM should be performed to improve the content.

12- The references part must be updated. See and use for instance the following recent papers

https://doi.org/10.1016/j.tca.2020.178747

https://doi.org/10.1016/j.cej.2020.128010

Reviewer 3 Report

The manuscript by Lisi and coworkers describes the “The Combined Effect of Ambient Conditions and Diluting Salt on the Degradation of Picric Acid: An In Situ DRIFT Study.” They presented this manuscript as technically sound and well-presented with the degradation of picric acid under different conditions. They explained the PA degradation using In Situ DRIFT spectroscopy and thermogravimetric analysis. I have one doubt that, in general, energetic compounds or organic compounds with unstable functional groups are not stable for a longer time. They easily undergo degradation or decomposition at ambient temperature. PA has three nitro groups which may easily release proton from the OH group when it is mixed with electron deficient cations. I do not think KBr has the ability to abstract the proton from OH group. Is there any effect of pre-heated PA at 80℃? Authors should consider adding a few experiments with pre-heated PA at 80℃ for up to 24 hours and 48 hours.

Another one, on Page 6, paragraph 3, “On the other hand, the formation of an amino group from a nitro group through the initial transfer of hydrogen was reported for trinitrotoluene [3] and can easily be assumed for PA as well”. Authors should discuss more details about the assumption of nitro conversion to the amino of PA.

Round 2

Reviewer 2 Report

I believe that the paper can be accepted in its current form